# Investigation on the Impact of Excitation Amplitude on AFM-TM Microcantilever Beam System’s Dynamic Characteristics and Implementation of an Equivalent Circuit

**DOI:** 10.3390/s24010107

**Published:** 2023-12-25

**Authors:** Peijie Song, Xiaojuan Li, Jianjun Cui, Kai Chen, Yandong Chu

**Affiliations:** 1School of Electrical Engineering. Lanzhou Jiaotong University, Lanzhou 730070, China; spj209@163.com; 2Gansu Institute of Metrology, Lanzhou 730050, China; 3Geometric Sciences Institute, National Institute of Metrology, Beijing 100013, China; ycuijj@163.com (J.C.);

**Keywords:** atomic force microscopy, nonlinear dynamic system, multistability, Multisim, equivalent circuit

## Abstract

Alterations in the dynamical properties of an atomic force microscope microcantilever beam system in tapping mode can appreciably impact its measurement precision. Understanding the influence mechanism of dynamic parameter changes on the system’s motion characteristics is vital to improve the accuracy of the atomic force microscope in tapping mode (AFM-TM). In this study, we categorize the mathematical model of the AFM-TM microcantilever beam system into systems 1 and 2 based on actual working conditions. Then, we analyze the alterations in the dynamic properties of both systems due to external excitation variations using bifurcation diagrams, phase trajectories, Lyapunov indices, and attraction domains. The numerical simulation results show that when the dimensionless external excitation *g* < 0.183, the motion state of system 2 is period 1. When *g* < 0.9, the motion state of system 1 is period 1 motion. Finally, we develop the equivalent circuit model of the AFM-TM microcantilever beam and perform related software simulations, along with practical circuit experiments. Our experimental results indicate that the constructed equivalent circuit can effectively analyze the dynamic characteristics of the AFM-TM microcantilever beam system in the presence of complex external environmental factors. It is observed that the practical circuit simulation attenuates high-frequency signals, resulting in a 31.4% reduction in excitation amplitude compared to numerical simulation results. This provides an essential theoretical foundation for selecting external excitation parameters for AFM-TM cantilever beams and offers a novel method for analyzing the dynamics of micro- and nanomechanical systems, as well as other nonlinear systems.

## 1. Introduction

The rapid evolution of electromechanical systems offers a significant opportunity for the extension of geometric measurement from macro to micro- and nano-scales [1]. Nonetheless, the advancements in measurement technology within the micro and nano-domains have revealed that certain dynamic characteristics of these mechanical systems cannot be fully elucidated by classical mechanics. Consequently, the impact of quantum mechanics on microelectromechanical systems (MEMS) is becoming increasingly prominent, and its mode of action has garnered widespread attention in recent years [2,3].

The atomic force microscope (AFM) is a prominent nanoscale measurement instrument. Its fundamental operating principle involves converting the van der Waals forces between atoms into displacement signals, which can be detected by the photoelectric detection system via the micro-cantilever beam system and the pressure change detection system [4,5,6,7]. By analyzing the alteration of the displacement signal, the morphological characteristics of the sample’s surface can be elucidated. AFM is extensively employed in the measurement and investigation of surface morphology in polymer materials, ceramics, biological cells, and other micro- and nano-materials [8,9,10,11].

The AFM-TM has progressively emerged as the most extensively employed AFM model, attributed to its exceptional measurement precision and minimal disruption to the surface topography of the tested sample. Balthazar and colleagues have discovered that the AFM-TM microcantilever exhibits intricate dynamic characteristics under specific parameter conditions, due to the interplay of quantum forces between the sample’s surface and the microcantilever’s tip [4]. The emergence of complex motion characteristics in microcantilever beams not only severely compromises the measurement accuracy of AFM-TM, substantially undermines its resolution, but also exacerbates the wear of the microcantilever probe’s tip and increases the risk of damage to the sample’s surface morphology [12,13,14]. In practical application scenarios, investigating the stable operation interval of the microcantilever beam system becomes crucial to avoid the impact of complex motion characteristics, such as high period doubling motion, on measurement accuracy. This has emerged as the central focus of research on the dynamic characteristics of AFM-TM microcantilever beam systems.

Before delving into the dynamical properties of the AFM-TM microcantilever system, it is imperative to perform mathematical modeling of the microcantilever system. Belardinelli et al. conducted an in-depth investigation of the mechanical structure of the AFM-TM microcantilever beam and characterized its mechanical model as a mass-spring-damping system [15]. Building upon this foundation, Ribeiro et al. further took into account the Casimir force and integrated it into the equivalent mathematical model to scrutinize the vibration characteristics [16].

The AFM-TM microcantilever beam presents a significant challenge in this study due to its intricate and delicate structure, as well as the stringent experimental requirements. Furthermore, the dynamic analysis process of nonlinear mechanical systems based on numerical simulation often relies on idealized assumptions, such as neglecting energy dissipation caused by coupled nonlinear impedance during vibration and disregarding changes in dynamic parameters due to thermal effects. Additionally, external complex environmental factors can also impact the system’s dynamic characteristics. Therefore, while numerical simulations provide valuable insights, their theoretical guidance may be limited. However, thanks to the rapid advancements in nonlinear circuits, mechatronics, and other related fields, a novel approach to validating the results of numerical analysis experiments has been developed [17,18,19,20,21,22,23,24,25,26,27,28,29,30], namely, equivalent circuit experiments. Marcondes et al. successfully validated the numerical analysis outcomes by constructing the equivalent circuit of the Watt centrifugal regulator system [31]. Therefore, in this study, we have developed an equivalent circuit system based on the mathematical model of the AFM-TM microcantilever beam system. This approach establishes a bridge between numerical analysis and the physical-mechanical system, thereby simplifying dynamics analysis while providing valuable guidance for modeling and simulation.

The organization of the complete paper is as follows: In Section 2, we present the mathematical model of the AFM-TM microcantilever developed by Ribeiro et al. In Section 3, subsystem 2 is isolated from the main system 1 based on actual working conditions, and the dynamic characteristics of systems 1 and 2 are analyzed under varying external excitation amplitudes using bifurcation diagrams, Lyapunov index diagrams, Poincare cross sections, phase trajectories, and attraction domains. The optimal range of external excitation amplitudes is determined to ensure stable motion of the respective system. In Section 4, we derive the corresponding state-space equation for the equivalent circuit based on that of system 1’s state-space equation. The accuracy of the circuit schematic is verified through Multisim simulation and error theory. Subsequently, we construct the physical circuit using this equivalent schematic and perform verification experiments. Finally, a comparative analysis between circuit simulation results and numerical simulation results is presented. A comprehensive summary is provided in Section 5.

## 2. Mathematical Model

### 2.1. Mathematical Model of System 1

The schematics of an AFM-TM are shown in Figure 1a,b [16]:

The motion equation of the cantilever beam system is expressed as:(1)$$m\overset{••}{x}+c_{1}\overset{•}{x}+c_{s}\overset{•}{x}+k_{1}x+k_{2}x^{3}=F_{vdM}(l_{0}+x)+F_{Cas}(l_{0}+x)+f\cos\omega t,$$

Define Equation (1) as system 1, where $c_{1}$ is the structural damping coefficient, $c_{s}$ is the equivalent extrusion film damping coefficient, $k_{1}$ is the equivalent linear stiffness, $k_{2}$ is the nonlinear equivalent stiffness, *k_at_* is the system coupling equivalent stiffness, f and ω are the amplitude and angular frequency of the harmonic driving force. Calculate Van der Waals forces [32,33] between the tip of a microcantilever beam and the sample,
(2)$$F_{vdM}(r)=\frac{AR}{180r^{8}}-\frac{BR}{6r^{2}},$$
where *A* is the Hamaker constant to the attractive potential, and *B* is the Hamaker constant to the repulsive potential. If the distance between the tip and the sample is $l_{0}$ when the cantilever beam is stationary, the distance between the tip and the sample is *r* when it is moving,
(3)$$r=l_{0}+x(t).$$

The Casimir force [34,35,36] is a compensating force after taking into account the hysteresis effect of the van der Waals force and is expressed as:(4)$$F_{Cas}(r)=\frac{\pi^{2}\hbar c_{l}}{240r^{4}},$$
ℏ is Planck’s constant and $c_{l}$ is the speed of light.

Define the balance distance parameter $l_{s}=(3/2){\left(2D\right)}^{1/3},$ $D=BR/(6k_{1}).$ Introduce dimensionless quantities:(5)$$\tau =\omega_{n}t,\;x_{1}=\frac{x}{l_{s}},x_{2}=\frac{\overset{•}{x}}{\omega_{n}l_{s}},a=\frac{l_{0}}{l_{s}},b=\frac{k_{1}}{k_{at}},c=\frac{k_{2}}{k_{at}}l_{s}^{2},d=\frac{4}{27},\;e=\frac{2a^{6}}{405l_{s}^{6}},p=\frac{16\mu_{eff}R^{7}}{m\omega_{n}l_{s}^{3}},\psi =\frac{h\pi^{2}c_{l}}{240},g=\frac{f}{k_{at}l_{s}},\Omega =\frac{\omega }{\omega_{n}},r=\frac{1}{Q},\;Q=\frac{m\omega_{n}}{c_{1}},\beta =\frac{\psi }{\omega_{n}^{2}l_{s}^{5}}.$$

The coefficient of effective viscosity is denoted by $\mu_{eff}$, the quality factor of the microcantilever beam is represented by Q, the first-order resonance frequency of the microcantilever beam is indicated by $\omega_{n}$, and R refers to the radius of the sphere after enlarging the tip of the microcantilever beam. After incorporating Formula (5) into Formula (1) and normalizing it, the dimensionless motion Equation (6) of the microcantilever beam system is derived.
(6)$${\overset{••}{x}}_{1}+r{\overset{•}{x}}_{1}+bx_{1}+cx_{1}^{3}=\frac{e}{{\left(a+x_{1}\right)}^{8}}-\frac{d}{{\left(a+x_{1}\right)}^{2}}-\frac{p}{{\left(a+x_{1}\right)}^{3}}{\overset{•}{x}}_{1}+\frac{\beta }{{\left(a+x_{1}\right)}^{4}}+g\cos(\Omega \tau ).$$

The state-space equation of the microcantilever system is
(7)$$\left\{\begin{array}{c} \overset{•}{x_{1}}=x_{2}\\ {\overset{•}{x}}_{2}=-rx_{2}-bx_{1}-cx_{1}^{3}+\frac{e}{{\left(a+x_{1}\right)}^{8}}-\frac{d}{{\left(a+x_{1}\right)}^{2}}-\frac{p}{{\left(a+x_{1}\right)}^{3}}x_{2}+\frac{\beta }{{\left(a+x_{1}\right)}^{4}}+g\cos(\Omega \tau ) \end{array}\right..$$

The conclusion can be drawn from (6) and (7) that the equation of motion for the cantilever beam system is a second-order nonlinear non-autonomous equation. The main physical parameters referenced in the numerical simulation below are tabulated in Table 1 [15,37].

### 2.2. Mathematical Model of System 2

In the absence of media interactions between the tip of the microcantilever beam and the sample under test, as well as between the tip and the sample, the microcantilever beam can be considered as the light beam mode depicted in Figure 2. The equation of motion is expressed as
(8)$$m\overset{••}{x}+c_{1}\overset{•}{x}+k_{1}x+k_{2}x^{3}=f\cos\omega t,$$

Define Equation (8) as system 2, bring $\tau =\omega_{n}t$ and the system coupling equivalent stiffness $k_{at}$ into Formula (9), and let the equilibrium distance $l_{D}=l_{s}$,
(9)μD=1Q,aD=k1kat,bD=k2lD2kat,fD=fkatlD,Ω=ωωn,x(τ)=x(t)lD,y(τ)=x•(t)ωnlD,
the state space equation of system 2 is
(10)$$\left\{\begin{array}{c} \overset{•}{x}=y\\ \overset{•}{y}=-\mu_{D}y-a_{D}x-b_{D}x^{3}+f_{D}\cos\Omega \tau \end{array}\right..$$

In the course of practical application, to preclude the detrimental effects triggered by the atomic force microscope’s micro-cantilever beam exceeding its operational range, the parameters of system 2 should be fine-tuned prior to the installation of the sample intended for testing. Activating the micro-cantilever beam is necessary to determine its effective travel. Only when the effective stroke remains within the designed range, and the motion characteristics of the micro-cantilever beam exhibit stability, allowing for the measurement of motion data, can the sample be added for testing.

## 3. The Influence of Excitation Amplitude on the Movement Properties of System 2 and System 1

As a crucial subsystem of system 1, a comprehensive examination of the bifurcation characteristics of system 2 can significantly contribute to unraveling its dynamic properties. This analysis offers robust support for the identification of complex system behaviors and the investigation of correlations between parameter modifications and motion characteristics.

### 3.1. System 2

Set the initial value $\left(x_{0},y_{0})=(0,0\right)$, when $\mu_{D}$ = 0.01, $a_{D}$ = 0.05, $b_{D}$ = 0.39, Ω = 1, with $f_{D}\in (0,1)$. Calculate the Lyapunov exponent *λ_i_* (*i* = 1, 2, 3) for each point within the corresponding interval. The bifurcation characteristics and Lyapunov exponent diagram of system 2 are presented in Figure 3:

As illustrated in Figure 3b, within the $f_{D}\in (0,\;1)$, no Lyapunov exponent surpasses 0, and (*λ*_1_, *λ*_2_, *λ*_3_) = (0, -, -). This indicates that, during this interval, the phase trajectory of system 2 ultimately converges to a limit cycle. To further explore the dynamic mechanism of system 2, we divide the bifurcation diagram of system 2 into three regions: I, II, and III as shown in Figure 3a. Subsequently, we analyze the attraction domains of the feature points BP_1_, BP_2_, and BP_3_ within these three regions and delve into the evolutional patterns of the attraction domains in each region.

#### 3.1.1. I—Regional Suction Basin Analysis

As illustrated in Figure 3a, BP_1_ represents the initial bifurcation point within the bifurcation diagram of system 2, at *f_D_* = 0.1833. The attractive domain within the space, characterized by *f_D_* = 0.1833, *x* ∈ (−0.5, 0.5), *y* ∈ (−0.5, 0.5), is depicted in Figure 4.

In the space delineated in Figure 4, five distinct attractor basins can be observed, namely: brown representing cycle 1’s attractor basin *Q*_1_, light green denoting cycle 2’s attractor basin, yellow indicating cycle 3’s attractor basin *Q*_3_, dark green corresponding to cycle 5’s attractor basin, and white representing cycle 7’s attractor basin. At *f*_D_ = 0.1833, the initial point (*x*_0_, *y*_0_) = (0, 0) is situated at the borderline of the attractor basins of *Q*_1_ and *Q*_3_, indicating the presence of attractor coexistence at the initial condition. To further scrutinize the bifurcation and attraction domain evolution of region I, we present the attraction domains for *f*_D_ < 0.1833 and *f*_D_ > 0.1833.

According to the analysis presented in Figure 5, the initial value point is situated within the suction basin *Q*_1_ of period 1 when *f_D_* is less than 0.1833. As *f_D_* increases, the area of *Q*_1_ in period 1, surrounded by period 3, progressively shrinks. When *f_D_* surpasses 1.833, *Q*_1_ is no longer capable of accommodating the initial value point, leading it to fall into the period 3 suction basin *Q*_3_. Further elevation of *f_D_* results in the emergence of a period 1 suction basin exterior to the spatial region enclosed by the period 3 suction basin. Based on Figure 5, it remains inconclusive whether the external and internal cycle 1 attractors represent the same attractor. At *f_D_* = 0.2300, the suction basin has permeated the marginal region of the original suction basin, thereby blurring its boundary. When *f_D_* = 0.2600, cycles 1, 3, and 4 are intertwined. Consequently, it can be speculated that as *f_D_* continues to rise within a finite interval, the *Q*_3_ region near the initial value point may stratify and merge with the adjacent period 1 suction basin. This potential stratification and infiltration could lead to the initial point traversing multiple suction basins due to small fluctuations in *f_D_*.

#### 3.1.2. II—Regional Suction Basin Analysis

Crossing into region II, an increase in *f_D_* furthermore leads to a sudden change in the bifurcation diagram at point BP_2_, where *f_D_* reaches 0.3736. This transformation generates a structure consisting of two discontinuous branches, with the lower branch connecting to Region I and the upper branch connecting to Region III. In the space range of *f_D_* = 0.3736, *x* ∈ (−0.5, 0.5), *y* ∈ (−0.5, 0.5), the configuration of the attraction domain is illustrated in Figure 6.

The attractor represented by brown, which is the period 1 attractor, occupies the largest spatial region. The period 4 attractor represented by indigo blue, the period 25 attractor represented by dark blue, and various other scattered attractors are infiltrated within the period 1 attractor. To analyze the cause of the jump at point BP_2_, an initial value point ($x_{0}^{'},\;$ $y_{0}^{'}$) = (0.4, −0.4) is selected with all other parameters remaining unchanged. The bifurcation diagram of the corresponding system 2 in the interval of *f_D_* ∈ (0, 1) is presented in Figure 7a.

In Figure 7a, the bifurcation plot experiences a discontinuity at *f*_D_ = 0.256. When combined with the analyses of Figure 5d and Figure 3, it can be deduced that the discrepancy between the bifurcation diagrams of Figure 3 and Figure 7 is due to the distinct nature of the period 1 suction basins *Q*_1_ and $Q_{1}^{'}$ in the lower right corner of Figure 5c,d. Specifically, at *f*_D_ = 0.3736, the *Q*_1_ suction basin begins to merge with the $Q_{1}^{'}$ suction basin, resulting in the bifurcation diagram of Region II in Figure 3 exhibiting a structure composed of an upper and lower branch. The lower branch represents the period-doubling attractor generated by the *Q*_1_ bifurcation, while the upper branch represents the period-doubling attractor generated by the $Q_{1}^{'}$ bifurcation. Consequently, it can be inferred that although the period 1 suction basin occupies a significant portion of the space in Figure 6, there are two distinct period 1 suction basins within this space, exhibiting interpenetrating and alternating winding patterns. Therefore, the bifurcation and motion characteristics of system 2 are found to be unstable.

Let us further investigate the evolution of the attraction domain in Region II. Under the conditions of *f_D_* = 0.4200, *x* ∈ (−0.05, 0.25), *y* ∈ (−0.2, 0.2), its attractive domain is depicted in Figure 8a. Under the conditions of *f_D_* = 0.4700, *x* ∈ (−0.5, 0.5), *y* ∈ (−0.5, 0.5), its attractive domain is illustrated in Figure 8b.

Figure 8a demonstrates the presence of three attractors in the space, namely, the brown attractor of period 1, the red attractor of period 2, and the green attractor of period 6. The initial point is situated within the period 6 suction basin and adjacent to the period 2 suction basin. In Figure 8b, only the brown period 1 attractor and the red period 2 attractor remain in the space. Based on Figure 3, it can be inferred that the period 4 suction basin represented by green in Figure 8a belongs to the lower branch bifurcation line in Region II. To determine the bifurcation line attribution of the period 2 suction basin represented by red in Figure 8a,b, the white marked point in Figure 8a is selected as the initial value point, with coordinates ($x_{0}^{'}$, $y_{0}^{'}$) = (−0.0011, 0.0172). At this point, the bifurcation diagram of system 2 varying with *f*_D_ is presented in Figure 7b. When *f*_D_ = 0.4200, the initial value point ($x_{0}^{''}$, $y_{0}^{''}$) is positioned within the period 2 suction basin, as denoted by the red marked point in Figure 7b. In this instance, the bifurcation point of period 2 belongs to the subsequent branch bifurcation line.

#### 3.1.3. III—Regional Suction Basin Analysis

According to Figure 3a, we observe that the bifurcation characteristics of System 2 are primarily determined by the upper branch bifurcation line upon entering Region III. By examining the suction basin at BP_3_ and its behavior across different values of *f*_D_ within region Ⅲ, we can discuss the motion characteristics of system 2 within region Ⅲ, as well as the evolution of the suction basin. Specifically, the *f*_D_ value corresponding to BP_3_ is 0.8100, and its attraction domain is depicted in Figure 9.

The period 1 attractor currently occupies the most significant space, yet numerous non-dominant attractors, including the period 2, 4, 5, 6, 19, and higher period number attractors, are scattered around the initial value point. Collectively, these attractors create a multitude of ribbon-like periodic attractors with blurred edges, interpenetrating the period 1 attractor. At this juncture, the motion characteristics of system 2 are unstable, and minor parameter adjustments or perturbations could result in unpredictable quasi-periodic motion. In summary, when the optimal value interval for the excitation amplitude is *f*_D_ ∈ (0, 0.1833), the motion characteristics of system 2 are manifested as period 1 motion, and the operation of system 2 is relatively stable, with a strong resistance to interference.

### 3.2. System 1

When the parameters *r* = 0.01, *a* = 1.6, *b* = 0.05, *c* = 0.39, *d* = 0.14, *e* = 0.0001, *p* = 0.0088, *β* = 0.02, Ω = 1, and the initial values (*x*_10_, *x*_20_) = (0, 0) are set, the resulting bifurcation diagram of system 1, with *g* ∈ (0, 0.183), is presented in Figure 10a.

From Figure 10a, it can be readily discerned that when *g* falls within the range of (0, 0.9), the motion state of system 1 is characterized by period 1, indicating a relatively stable operation.

The motion characteristics of system 1 within the red dotted box are of particular concern. Near a threshold value of *g* = 0.09660, system 1 undergoes a delayed bifurcation, resulting in the emergence of an unstable period 2 orbit in the intercell vicinity. Upon leaving this interval, system 1 manifests as a stable periodic 2 orbit. When *g* assumes values of 0.09640, 0.09660, and 0.09680, the corresponding phase trajectories are depicted in Figure 10b–d, respectively. The change values of these three phase trajectory parameters are identical, all being 0.002. Within this small range of variations, the phase trajectory of g exhibits two crucial turns: when *g* = 0.09640, the phase trajectory remains stable, displaying a small left and large right pattern; when *g* = 0.09660, the phase trajectory becomes unstable despite the graph being smaller on the left and larger on the right. When *g* = 0.09680, the phase trajectory stabilizes and the graph undergoes a transition to the larger left and smaller right. The rationale behind this alteration can be more intuitively understood through Figure 11: when *g* < 0.09660, the initial value point (*x*_10_, *x*_20_) is situated within the *Q*_2_ suction basin, resulting in the phase trajectory being smaller on the left and larger on the right. As the attraction domain evolves, when g > 0.09660, the initial value point enters the *Q*_2_ attraction basin, leading to the phase trajectory becoming larger on the left and smaller on the right. The white ribbon suction basin is characterized by an unstable suction basin.

## 4. Simulation of an Equivalent Circuit for the AFM-TM Microcantilever Beam System

### 4.1. Equivalent Circuit Design

Before fabricating the corresponding chaotic circuit, it is imperative to undergo the circuit normalization process in order to facilitate the translation of mathematical language into circuit language. In this study, we have selected a normalized resistance value of *R_n_* = 10 kΩ, a normalized capacitance value of *C_n_* = 0.1 μF, and a normalized time constant *t_n_*, which is derived from the product of the normalized resistance and capacitance, i.e., *R_n_C_n_*, equaling 1 ms. According to Equation (7), the schematic diagram of the equivalent circuit is illustrated in Figure 12.
(11)$$u_{x_{e}}=\frac{1}{R_{3}C_{1}}\int \frac{R_{1}}{R_{2}}u_{y_{e}}dt_{e},$$
(12)$$u_{A_{e}}=-R_{19}(\frac{u_{y_{e}}}{R_{6}}+\frac{u_{x_{e}}}{R_{4}}),$$
(13)uBe=−R8uxe3R5+uV0R7R312(uxeR29+uV1R30)2,
(14)uCe=−R11uV0R9R314(uxeR29+uV1R30)4+uV02R10R318(uxeR29+uV1R30)8,
(15)uDe=−R17R12C2R15·duV4dte−R33uV0uyeR16R32R313(uxeR9+uV1R30)3,
(16)$$u_{y_{e}}=\frac{R_{27}}{R_{28}C_{3}}\int \left[\frac{R_{18}R_{23}}{R_{24}R_{25}}(\frac{u_{A_{e}}}{R_{13}}+\frac{u_{B_{e}}}{R_{14}})-\frac{R_{22}}{R_{26}}(\frac{u_{C_{e}}}{R_{20}}+\frac{u_{D_{e}}}{R_{21}})\right]dt_{e},$$

The voltage input-output relationship of each node in Figure 12 is as follows:

Let
(17)$$r_{e}=\frac{R_{n}}{R_{6}},b_{e}=\frac{R_{n}}{R_{4}},c_{e}=\frac{R_{n}}{R_{5}},d_{e}=\frac{R_{n}}{R_{7}},\beta_{e}=\frac{R_{n}}{R_{9}},e_{e}=\frac{R_{n}}{R_{10}},p_{e}=\frac{R_{n}}{R_{16}},$$
in addition to *R*_4_, *R*_5_, *R*_6_, *R*_7_, *R*_9_, *R*_10_, and *R*_16_, the resistance values of the remaining resistors are set to 10 kΩ, while their capacitance values are established as the normalized capacitance value of 0.1 μF. By substituting the Formula (17) and the values of other resistors into the Formulas (11)–(16), the state space equation of the equivalent circuit is derived after simplification and arrangement:(18)$$\left\{\begin{array}{c} \frac{du_{x_{e}}}{dt_{e}}=\frac{u_{y_{e}}}{R_{3}C_{1}}\\ \frac{du_{y_{e}}}{dt_{e}}=\frac{1}{R_{28}C_{3}}\left[-r_{e}u_{y_{e}}-b_{e}u_{x_{e}}-c_{e}u_{x_{e}}^{3}-\frac{u_{V_{0}}}{{\left(u_{x_{e}}+u_{V_{1}}\right)}^{2}}+\frac{\beta_{e}u_{V_{0}}}{{\left(u_{x_{e}}+u_{V_{1}}\right)}^{4}}+\frac{e_{e}u_{V_{0}}^{2}}{{\left(u_{x_{e}}+u_{V_{1}}\right)}^{8}}-\frac{p_{e}u_{V_{0}}u_{y_{e}}}{{\left(u_{x_{e}}+u_{V_{1}}\right)}^{3}}\right]+\frac{du_{V_{4}}}{dt_{e}} \end{array}\right.,$$
meanwhile, let
(19)$$u_{V_{0}}=1\;V,\;u_{V_{1}}=1.6\;V,\;u_{V_{4}}=h\sin(f_{e}t_{e}),\;$$
where the term *h* denotes the amplitude of the external excitation voltage, while *f_e_* represents the frequency of the external excitation voltage. In this particular instance, the state space Equation (18) of the equivalent circuit is identical to the state space Equation (7) of the AFM-TM microcantilever system.

### 4.2. Simulation of AFM-TM Equivalent Circuit Using Software

Let *R*_4_ = 200.0 kΩ, *R*_5_ = 25.5 kΩ, *R*_6_ = 100.0 kΩ, *R*_7_ = 71.5 kΩ, *R*_9_ = 500.0 kΩ, *R*_10_ = 100 MΩ, and *R*_16_ = 1.13 MΩ, according to Formula (17), the corresponding *b_e_* = 0.05, *c_e_* = 0.39, *r_e_* = 0.1, *d_e_* = 0.14, *β_e_* = 0.02, *e_e_* = 0.0001, and *p_e_* = 0.0088 can be obtained. Additionally, the output excitation amplitude of the signal generator is set equal to the external excitation amplitude of the numerical simulation, and the frequency value of the equivalent circuit’s external excitation is calculated according to Formula (20)
(20)$$f_{e}=\frac{\Omega }{2\pi R_{n}C_{n}},\;$$
according to Equation (20), the theoretical value of the external excitation voltage frequency *f_e_* for the equivalent circuit is 159.2 Hz, which corresponds to the first-order resonance frequency of the mechanical system.

In the course of circuit software simulation, the subsequent circuit and environmental simulation conditions can influence the normalization time, resulting in a biased simulation outcome. Consequently, when the amplitude of the external excitation voltage alters, the deviation value correspondingly changes. To ensure the accuracy of the simulation results, the voltage frequency should be fine-tuned concurrently following each voltage adjustment. To calculate the deviation value, a reference point is initially established, followed by adjusting the frequency of the signal generator to minimize the relative error between the circuit software simulation phase track value and the numerical simulation phase track value. Throughout this process, the actual frequency *f_e_* of the signal generator is documented. The maximum positive direction value of the longitudinal coordinate of the phase trajectory is designated as the reference point, with the value *x*_2max_ obtained from numerical simulation defined as the standard value, and the value *y^’^_e_*_max_ (dimensionless quantity) derived from the equivalent circuit software simulation considered as the simulation value. The relative error of the actual frequency value versus the theoretical value is further defined as *e_f_*.

When the value of *h* is set to 0.06 V, 0.10 V, 0.14 V, and 0.22 V, respectively, the comparison between the numerical simulation phase trajectory and the circuit software simulation phase trajectory is presented in Figure 13:

By meticulously examining Table 2, we discern that the relative error associated with circuit software simulation is minimal when the AFM-TM microcantilever beam system demonstrates periodic or quasi-periodic motion characteristics. Conversely, when the motion characteristics of the circuit display chaotic behavior, the relative error of the simulation becomes more pronounced. Within the specified amplitude range, the composite error of the circuit software simulation remains below 3% in comparison to the numerical simulation. After conducting five iterations of nonlinear fitting on the relative error column data presented in Table 2, the obtained fitting results are depicted in Figure 14.
(21)$$\tilde{e_{1}}=\;333972\times h^{'}{}^{5}\;-\;329052\times h^{'}{}^{4}+120170\times h^{'}{}^{3}\;-\;19946\times h^{'}{}^{2}+1488.8\times h^{'}\;-\;39.307,$$
which represents the trend line of the quintic fitting of the relative error of the ordinate,
(22)$$\tilde{e_{f}}=\;-16589\times h^{'}{}^{5}+11373\times h^{'}{}^{4}\;-\;1754.7\times h^{'}{}^{3}\;-\;300.59\times h^{'}{}^{2}+106.33\times h^{'}\;-\;6.4191,$$
which represents the quintic fitting trend line of the frequency relative error. It is calculated that the fitting degree $\tilde{e_{1}}$ of trend line $R_{e_{1}}^{2}=0.927$; the fit degree of trend line $\tilde{e_{f}}$ is $R_{e_{f}}^{2}=0.979$.

### 4.3. PCB Simulation Experiment

The PCB (Printed Circuit Board) was designed in accordance with the circuit schematic diagram, as illustrated by the experimental PCB in Figure 15, the operational amplifier chip model employed in the circuit is AD743JN, while the multiplication/voltage division chip model is AD734ANZ.

The models of instruments and some indicators used in the experiment are as follows: Waveform Generator: KEYSIGHT 33500B Series (total harmonic distortion: 0.008%); Oscilloscope: Tektronix MDO3012 (vertical resolution 8 bit, high resolution 11 bit); DC power supply: 4NIC-X264 (voltage accuracy: less than 1.0% at 15 V). The schematic diagram of the experimental setup is presented in Figure 16:

The most significant phase trajectory feature of system 1 is the trajectory within the range of the red dashed line in Figure 10a. To identify the corresponding image, we adjust the external excitation amplitude and frequency. When the voltage, *h*, is set at 0.14 V and 0.15 V, intriguing modifications can be observed in the phase trajectory, as illustrated in Figure 17a,b. Alterations in the small excitation amplitude result in the simulated phase trajectory undergoing a reversal, suggesting that the system is influenced by an unstable equilibrium point, as analyzed in Figure 10c,d. As illustrated in Figure 17c, when *h* = 0.34 V, the system exhibits a distinct and stable chaotic motion state, corresponding to the patterns observed in Figure 13d,h.

By comparing the numerical simulation of the AFM-TM microcantilever beam system with the actual circuit simulation, we find that the actual circuit simulation can describe the evolution of the system’s motion characteristics when the external excitation amplitude varies. However, due to the presence of numerous interference factors in the actual environment, the actual circuit simulation results struggle to accurately present the details of each motion state compared to the numerical simulation results. For instance, as illustrated in Figure 17a,b, the unstable equilibrium point consistently influences the phase trajectory, resulting in a relatively pale-hued phase trajectory at the periphery of the 2-period phase trajectory. Furthermore, the chaotic motion phase trajectory depicted in Figure 17c exhibits a higher degree of chaos compared to the numerical and software simulations. Nonetheless, it is the introduction of these interference factors that causes the initial value of the system to fluctuate slightly at each time point during the actual circuit simulation process, enabling direct observation of the influence of the unstable equilibrium point on the system. Furthermore, there are discrepancies between the actual circuit and the ideal condition, leading to significant energy loss, as illustrated in Figure 17a, where the loss of the external excitation amplitude reaches as high as 31.4%.

## 5. Discussion and Conclusions

From our investigation, we can derive the following crucial findings: Firstly, alterations in the amplitude of external excitation result in a notable complexity of the motion attributes in the AFM-TM microcantilever beam system. Secondly, employing the AFM-TM cantilever system as a nonlinear circuit facilitates the study of the effects of dynamic parameters (including, but not limited to, external excitation amplitude, equivalent stiffness, equivalent damping, and Hamaker coefficient etc.) on the motion characteristics of the mechanical system. Lastly, it is imperative to note that discrepancies exist between the experimental outcomes of the actual circuit and the numerical simulations, which unravel the influence mechanism of complex external environments on the system motion characteristics under suboptimal conditions.

The present study analyzes the evolution law of dynamic characteristics in an AFM-TM microcantilever beam system caused by variations in external excitation amplitude. It is noteworthy that external excitation frequency, equivalent stiffness, and equivalent damping are also common dynamic parameters in the equations governing both system 1 and system 2 dynamics. The research methodology employed herein can be utilized to analyze the influence mechanism of changes in other dynamic parameters on the motion characteristics of a microcantilever beam system under specific conditions. This will provide a theoretical reference for optimizing the design of dynamic parameters in AFM-TM microcantilever beam systems, fault detection in AFM-TMs, and enhancing measurement accuracy.

This study further enhances the interconnection between numerical analysis and actual mechanical system analysis by employing circuit simulation. In instances where the mechanical system is constrained by technical limitations, rendering it unable to be experimentally analyzed, but is significantly responsive to changes in the external environment, the application of the circuit simulation method becomes a prudent choice.

Due to the disparities between the actual equivalent circuit system and mechanical system, as compared to the idealized integer mathematical system, there exist transmission efficiency issues and energy losses. However, employing a fractional order model enables a better depiction of non-integer order response and memory effects in real systems. Therefore, our future research direction entails constructing a fractional order model for AFM-TM and subsequently aligning it with its equivalent circuitry to elucidate the underlying mechanism behind the energy losses in said circuit.

## Figures and Tables

**Figure 1 sensors-24-00107-f001:**
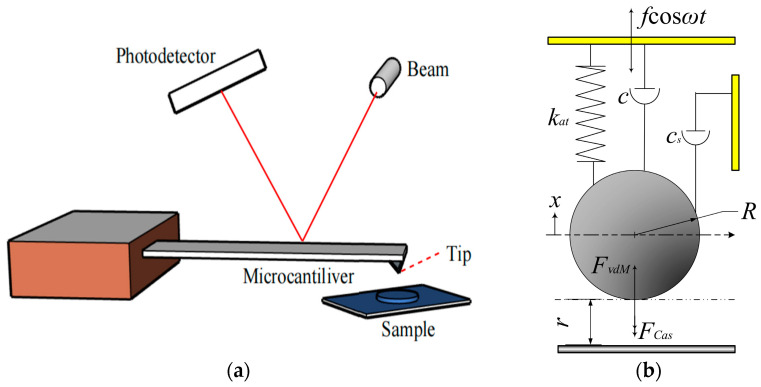
The schematic diagrams of the AFM-TM: (**a**) Microcantilever schematic diagram. (**b**) Physical model of the tip represented by a mass-spring-damper system.

**Figure 2 sensors-24-00107-f002:**
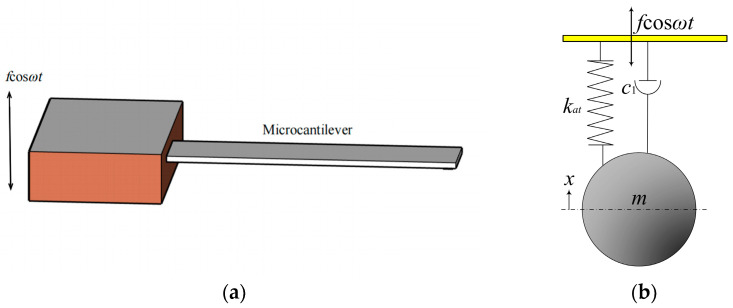
Light beam mode of a microcantilever beam. (**a**) Microcantilever schematic diagram. (**b**) Physical model of the tip represented by a mass-spring-damper system.

**Figure 3 sensors-24-00107-f003:**
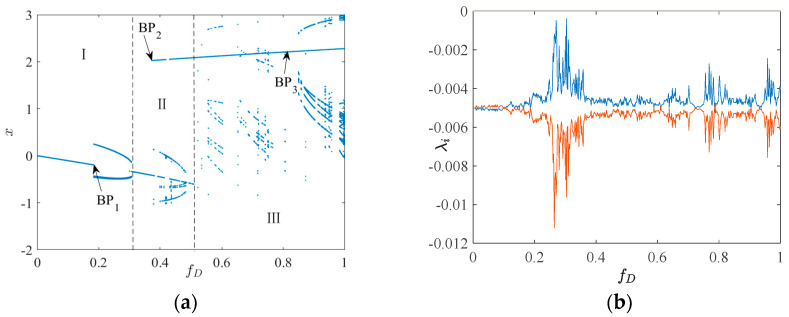
Bifurcation characteristic and Lyapunov indexes diagram of system 2: (**a**) bifurcation diagram; (**b**) Lyapunov indexes diagram.

**Figure 4 sensors-24-00107-f004:**
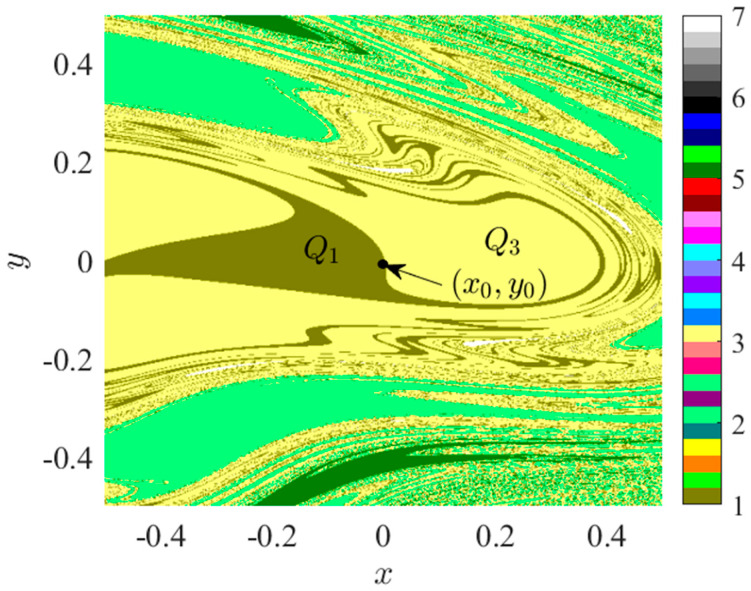
Attraction domain, *f_D_* = 0.1833.

**Figure 5 sensors-24-00107-f005:**
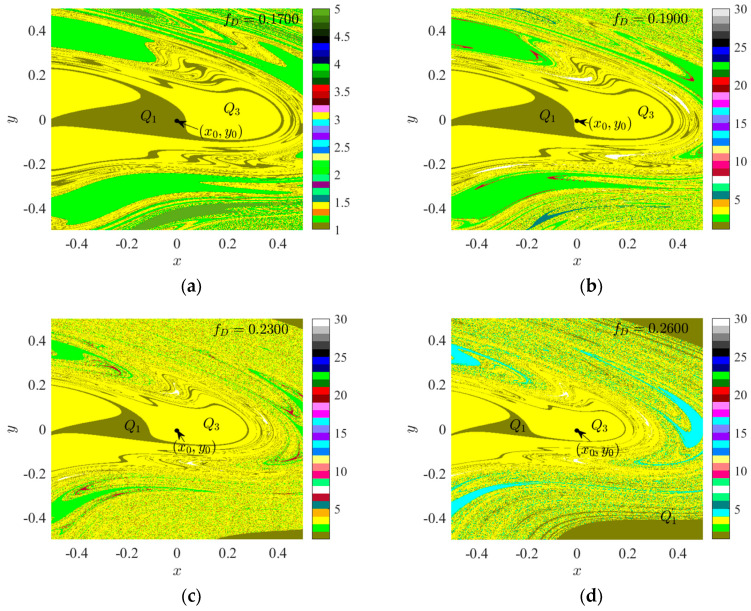
Attraction domain: (**a**) *f_D_* = 0.1700; (**b**) *f_D_* = 0.1900; (**c**) *f_D_* = 0.2300; (**d**) *f_D_* = 0.2600.

**Figure 6 sensors-24-00107-f006:**
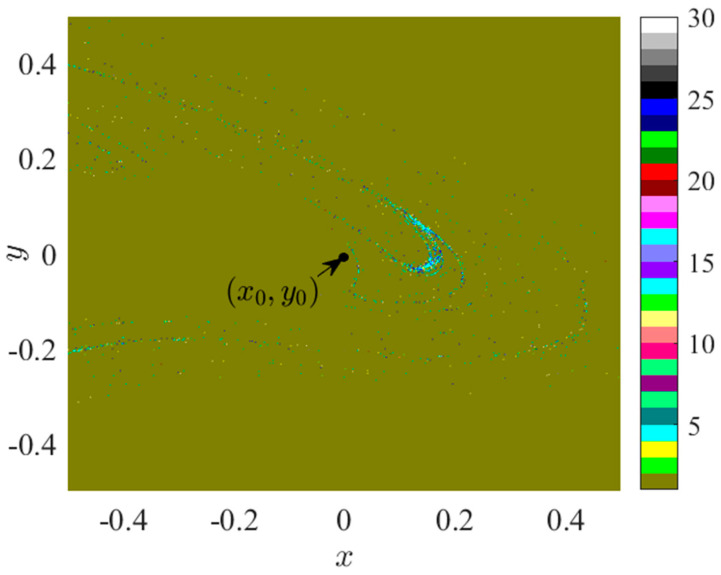
Attraction domain, *f_D_* = 0.2300.

**Figure 7 sensors-24-00107-f007:**
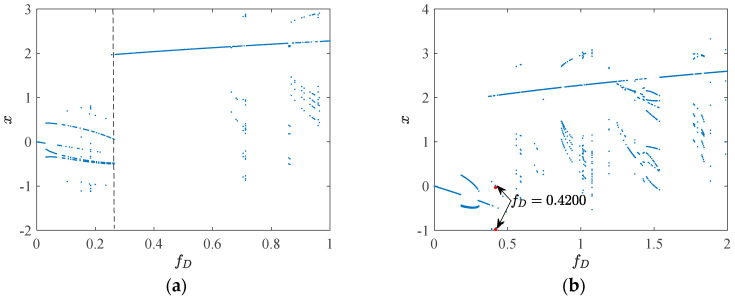
Bifurcation diagram for system 2: (**a**) Initial value ($x_{0}^{'}$, $y_{0}^{'}$) = (0.4, −0.4); (**b**) Initial value ($x_{0}^{''},$ $y_{0}^{''}$) = (−0.0011, 0.0172).

**Figure 8 sensors-24-00107-f008:**
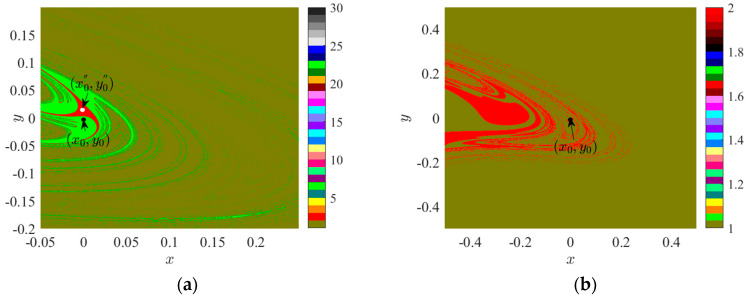
Attraction domain: (**a**) *f_D_* = 0.4200; (**b**) *f_D_* = 0.4700.

**Figure 9 sensors-24-00107-f009:**
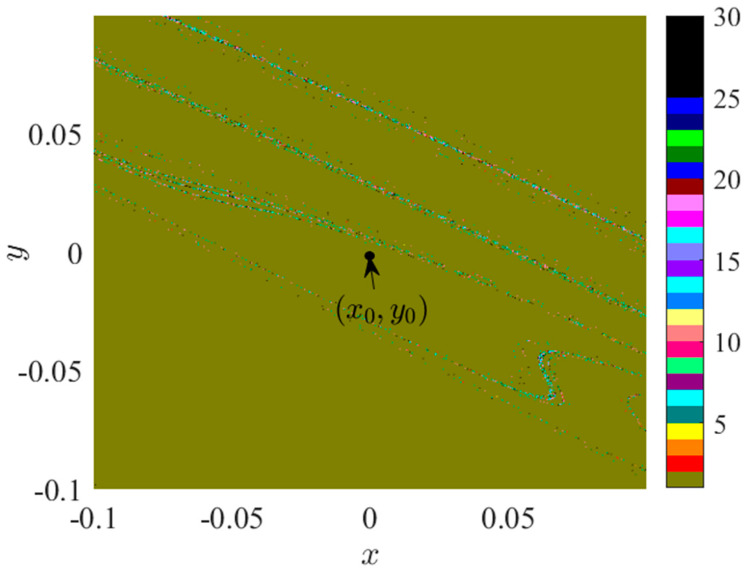
Attraction domain, *f_D_* = 0.8100.

**Figure 10 sensors-24-00107-f010:**
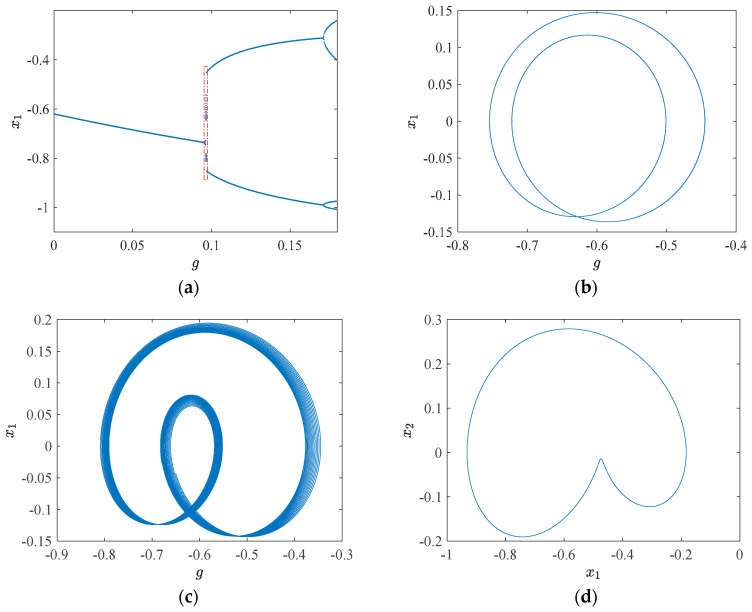
Bifurcation diagram and phase trajectories: (**a**) bifurcation diagram, 0 ˂ *g* ˂ 0.183; (**b**) phase trajectory, *g* = 0.09640; (**c**) phase trajectory, *g* = 0.09660;(**d**) phase trajectory, *g* = 0.09680.

**Figure 11 sensors-24-00107-f011:**
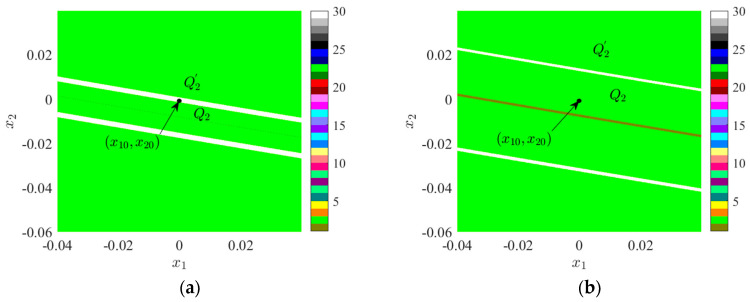
Attraction domain: (**a**) *g* = 0.9660; (**b**) *g* = 0.9640.

**Figure 12 sensors-24-00107-f012:**
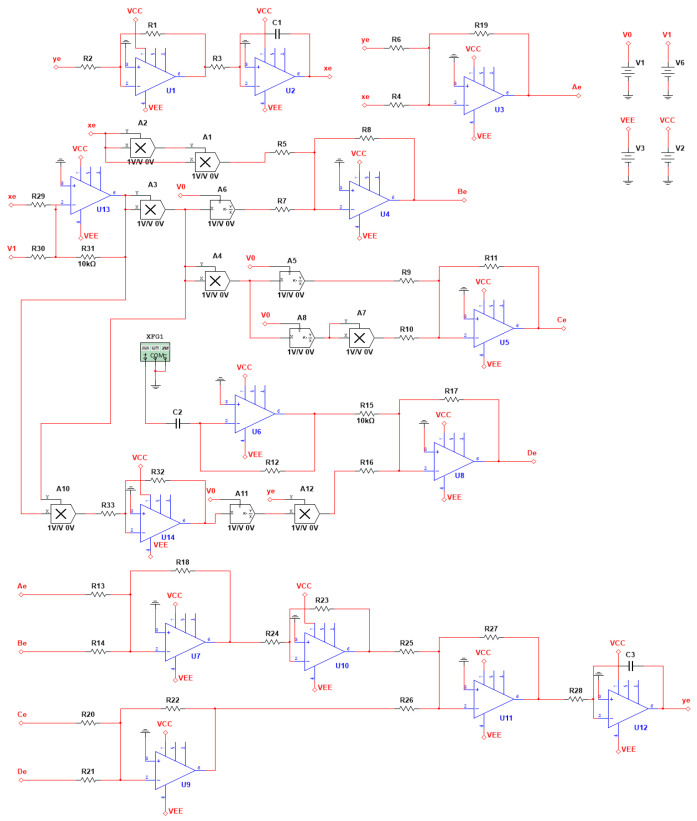
AFM-TM microcantilever beam system equivalent circuit schematic diagram.

**Figure 13 sensors-24-00107-f013:**
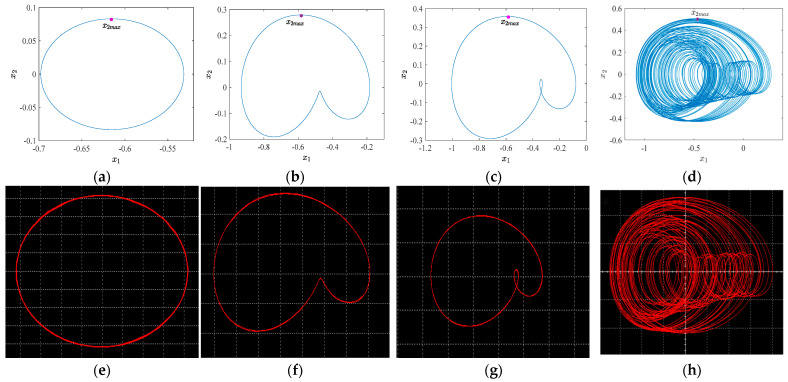
Phase trajectories: (**a**) numerical simulation, *f_D_* = 0.06; (**b**) numerical simulation, *f_D_* = 0.10; (**c**) numerical simulation, *f_D_* = 0.14; (**d**) numerical simulation, *f_D_* = 0.22; (**e**) circuit simulation, *h* = 0.06 V; (**f**) circuit simulation, *h* = 0.10 V; (**g**) circuit simulation, *h* = 0.14 V; (**h**) circuit simulation, *h* = 0.22 V.

**Figure 14 sensors-24-00107-f014:**
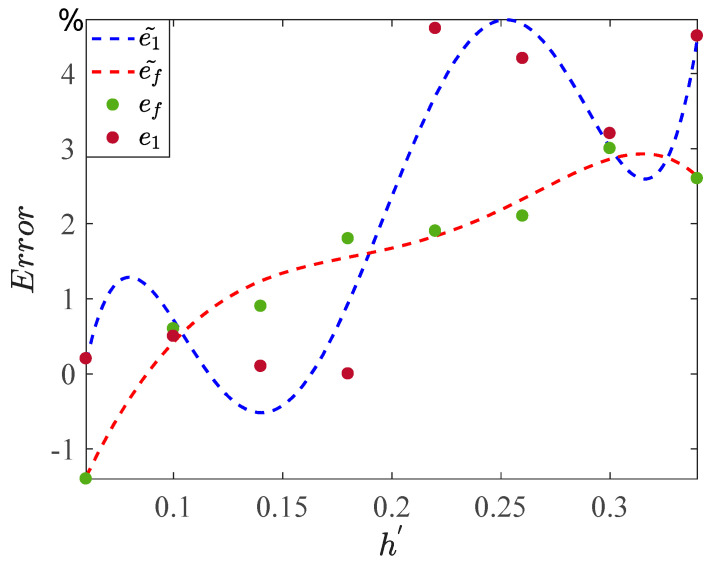
Relative error data points and fitted graphs.

**Figure 15 sensors-24-00107-f015:**
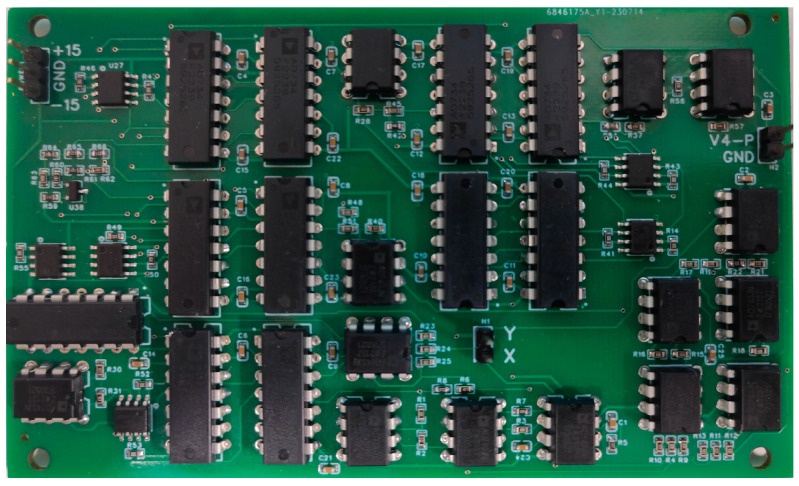
The Implementation of Experimental PCB.

**Figure 16 sensors-24-00107-f016:**
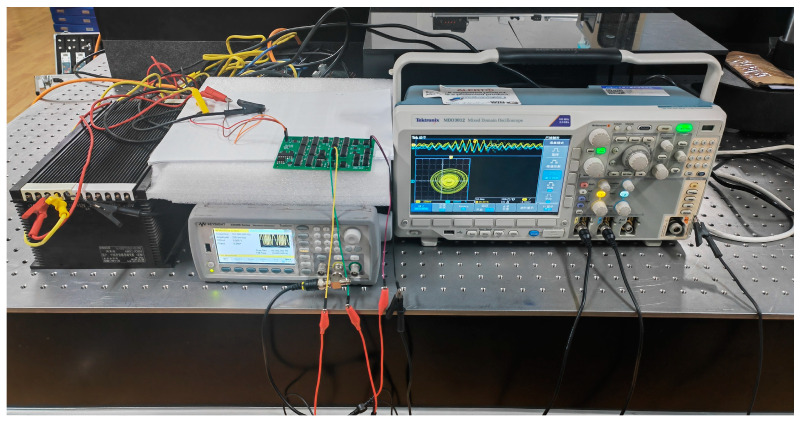
The schematic diagram of the experimental setup.

**Figure 17 sensors-24-00107-f017:**
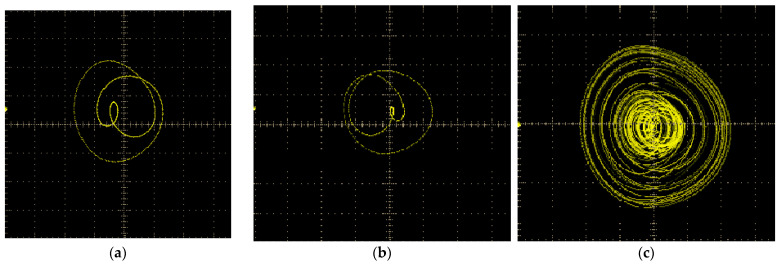
Phase trajectories obtained by actual circuit simulation: (**a**) *h* = 0.14 V; (**b**) *h* = 0.15 V; (**c**) *h* = 0.34 V.

**Table 1 sensors-24-00107-t001:** The main physical parameters of the AFM-TM microcantilever beam system.

Description	Value
Length of the microcantilever beam	449 μm
Width of the microcantilever beam	46 μm
Thickness of the microcantilever beam	1.7 μm
Radius of the tip	0.15 μm
Material density of the microcantilever beam	2330 kg/m^3^
Elastic modulus of materials used in microcantilever beams	176 GPa
Equivalent stiffness of system coupling	9.8 N/m
The first-order resonance frequency of the microcantilever beam	16,059 Hz
Quality factor of the microcantilever beam	100
Hamaker constant (attractive)	1.3596 × 10^−70^ J·m^6^
Hamaker constant (repulsive)	1.865 × 10^−19^ J

**Table 2 sensors-24-00107-t002:** Reference point experimental data and relative error.

*h* ^’^	*y* ^’^ _*e*max_	*x* _2max_	*f^’^* _e_	*f_e_*	*e*_1_ (%)	*e_f_* (%)
0.06	0.0833	0.0831	157	159.2	+0.2	−1.4
0.10	0.2803	0.279	160.2		+0.5	+0.6
0.14	0.3580	0.3576	160.7		+0.1	+0.9
0.18	0.4202	0.4204	162.1		0.0	+1.8
0.22	0.5302	0.5067	162.2		+4.6	+1.9
0.26	0.5966	0.5726	162.6		+4.2	+2.1
0.30	0.6605	0.6398	163.9		+3.2	+3.0
0.34	0.7102	0.6799	163.3		+4.5	+2.6

*h^‘^* is the dimensionless quantity of the external excitation voltage amplitude *h*.

## Data Availability

Data are contained within the article.

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
