# Peer review of "Investigation on the Impact of Excitation Amplitude on AFM-TM Microcantilever Beam System’s Dynamic Characteristics and Implementation of an Equivalent Circuit"

_sensors, 2023, doi:10.3390/s24010107_

Round 1
Reviewer 1 Report
Comments and Suggestions for Authors
Full Title: Investigation on the Impact of Excitation Amplitude on AFM- 2 TM Microcantilever Beam System's Dynamic Characteristics 3 and Implementation of an Equivalent Circuit
The above manuscript study the mathematical model of AFM-TM microcantilever beam system into systems 1 and 2 based on actual working conditions. then analyze the alterations in the dynamic properties of both systems due to external excitation variations using bifurcation diagrams, phase trajectories, Lyapunov indices, and attraction domains.. The abstract need covers some results that mentioned in paper, to be better. The introduction is need to be more comprehensive and the experimental work is good discussion. Some comments could be summarized as follows:
1- In the abstract should be rewritten, to refer on the induced changes of beam on microstructure, mechanical properties with some data.
2- The Keywords should be revised.
3- The objectives may be need to be clearer – need to modify the objective and the novelty of the work in the last paragraph of the introduction part?
4- Why author has chosen these parameters of the AFM-TM microcantilever beam system.? Give the practical reason for selecting these values?
5- Results are interesting, but should to be more discussion and comparison with other works?
6- Tables 1: maybe is better if the authors put the results of uncertainty?
7- How to measure the sample thickness?
8- The conclusions should to be more directed toward the application of beam in modifying the microstructure, mechanical properties.
9- You should to update the refs, some refs are old?
Reviewer 2 Report
Comments and Suggestions for Authors
The authors investigated the impact of excitation amplitude on AFM-TM microcantilever beam system's dynamic characteristics. And they develop the equivalent circuit model of the AFM-TM microcantilever beam and perform related software simulations, along with practical circuit experiments. Before publcation, the authors should address following issues.
1. Van der Waals => van der Waals
2. In introduction, the authors should express clearly about thier own work compared with previous work for originality, what the contribution the authors do in this field, why the autors did this work.
3. make clear in the figure 1(b) for k1, k1, kat, Fvdm, Fcas.
4. Define clearly system 2 and systme 1 before arguing them. Better to add some figure of descript system 1 and system 2 for imagination.
Comments on the Quality of English LanguageCheck some grammeric issues.
Round 2
Reviewer 2 Report
Comments and Suggestions for Authors
I recommend to be published.